# Dam Siting: A Review

**Yang Wang** [1,2]**, Yongzhong Tian** [1,2,]*** and Yan Cao** [1,2]

1   Chongqing Jinfo Mountain Karst Ecosystem National Observation and Research Station,
    School of Geo-Graphical Sciences, Southwest University, Chongqing 400715, China;
    wy970729@email.swu.edu.cn (Y.W.); cy19730503@email.swu.edu.cn (Y.C.)
2   Daotian Science and Technology Limited Company, Chongqing 400700, China
*   Correspondence: tyzlf@swu.edu.cn

**Abstract:** Dams can effectively regulate the spatial and temporal distribution of water resources, where the rationality of dam siting determines whether the role of dams can be effectively performed. This paper reviews the research literature on dam siting in the past 20 years, discusses the methods used for dam siting, focuses on the factors influencing dam siting, and assesses the impact of different dam functions on siting factors. The results show the following: (1) Existing siting methods can be categorized into three types—namely, GIS/RS-based siting, MCDM- and MCDM-GIS-based siting, and machine learning-based siting. GIS/RS emphasizes the ability to capture and analyze data, MCDM has the advantage of weighing the importance of the relationship between multiple factors, and machine learning methods have a strong ability to learn and process complex data. (2) Site selection factors vary greatly, depending on the function of the dam. For dams with irrigation and water supply as the main purpose, the site selection is more focused on the evaluation of water quality. For dams with power generation as the main purpose, the hydrological factors characterizing the power generation potential are the most important. For dams with flood control as the main purpose, the topography and geological conditions are more important. (3) The integration of different siting methods and the siting of new functional dams in the existing research is not sufficient. Future research should focus on the integration of different methods and disciplines, in order to explore the siting of new types of dams.

**Keywords:** dam siting; multi-criteria decision-making; geographic information systems; machine learning; siting factors

## 1. Introduction

Water is a basic human need [1], playing important roles in facilitating geophysical cycles [2], regulating microclimates and runoff cycles [3–5], and sustaining the life activities of the Earth's organisms [6,7]. Dams, on the other hand, regulate the hydrological environment of small areas at a small scale [8]. Dams are man-made structures or naturally occurring barriers that span rivers and raise water levels by controlling or impeding the flow of water. They provide effective regulation of the spatial distribution pattern of water resources [9], for purposes of soil and water conservation, water supply, irrigation, aquaculture, flood control, and power generation [10–15]. There are 58,713 registered dams in the world [16]. The economic value of dams far outweighs their disadvantages and costs, and they play significant roles in regulating the distribution of water resources and balancing water systems and ecosystems [17].

Dams are the key for hydraulic projects, but not all dam construction processes are based on a scientific and systematic approach to decision making. Due to anthropogenic and political factors, the neglect of the technical aspects of the problem is still present [18,19]. Reasonable siting solutions consider the balance between ecology and energy [20], reducing the associated damage to the environment [21]; poor siting can cause negative impacts, such as the risk of erosion leading to mudslides and landslides [22], serious impacts on runoff

and sedimentation processes [23], and low or negative economic benefits [24]. Therefore, studying the spatial distribution of reservoirs and making decisions on dam siting are key steps in water resource management.

In the light of the scientific literature, many researchers have analyzed the optimal location for dam construction. These studies have specific factors to determine the appropriate location and show variability in different purposes of dams, for example: irrigation, power generation, water supply, and flood control. A search of English literature in existing databases (excluding other languages) shows that there are many research results on dam siting, but few review papers are available, making it difficult to grasp the progress of the existing research and the future direction of development in this field.

Sustainable development is an important global issue, in which the development of clean energy makes an essential contribution. Existing studies include systematic reviews of wind and solar power plant siting [25–27], as well as a review of hydroelectric plant siting [28], which similarly involve the selection and trade-off of a large number of factors. Hydroelectric power generation is one of the important uses of dams, and a comprehensive review on dam siting is highly informative. Meanwhile, dam siting can provide a strong support for future systematic reviews of hydraulic power plant siting. In this review, the existing research results on dam siting were analyzed and discussed in terms of three aspects—siting methods, siting decision factors, and the influence of use and siting factors—with the intention of providing more systematic and scientific theoretical support for future dam siting projects.

## 2. Review Approach

### 2.1. Data Collection

A systematic literature search of peer-reviewed publications in Web of Science, Scopus, and Google Scholar databases was conducted to understand dam siting methods and the influence of factors on the siting process. The search terms were ("dam" or "reservoir" or "retention" or "basin "or "detention basin") and ("site selection" or "sitting"), searched between 2000 and 2020. A total of 1254 results were found, with 748 results duplicated by different databases. There were 358 search results, some of which had weak or lacking relevance to the topic; for example, in the search, there were papers on beaver dam building behavior [29], animal and plant habitats and reproduction [30], and solar/wind power plant site selection [31,32], which did not have applicability of review or statistical correlation. In order to be highly relevant to the research topic, the results were checked for accuracy and topic deviation, and the 148 articles obtained provided a qualitative analysis of dam siting methods. After a full-text review, a sample of 25 papers was selected for quantitative analysis of the selection of different types of factors in dam siting, and how dam use affects factor selection. We drew a flowchart (Figure 1) to illustrate the data collection strategy, and used Table 1 to illustrate the search conditions and content.

**Table 1.** Key elements in data collection.

| Description | Conditions/Contents |
| --- | --- |
| databases | Web of Science, Scopus, and Google Scholar |
| search equation | ("dam" or "reservoir" or "retention" or "basin" or "detention basin") and ("site selection" or "siting") |
| last search date | March 2021 |

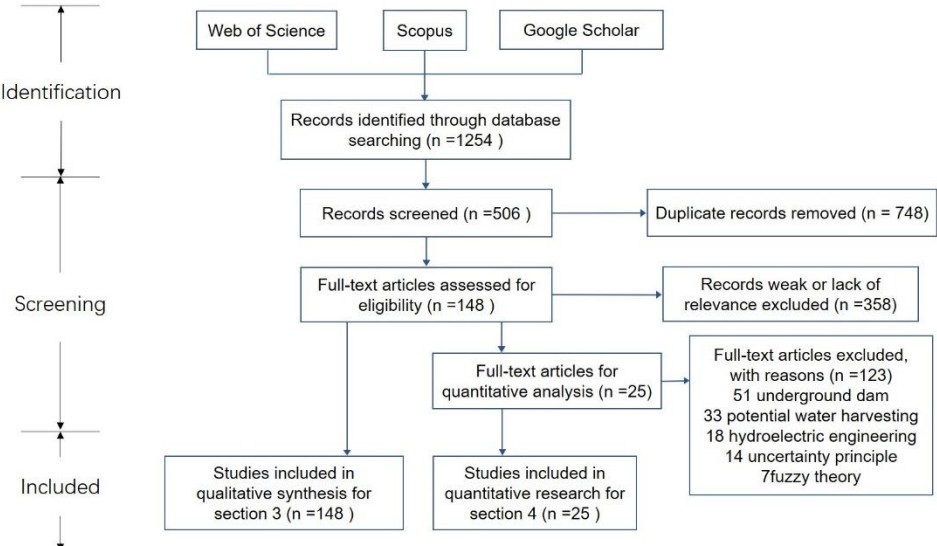

**Figure 1.** Flowchart for paper selection process.

### 2.2. Taxonomy

The siting methods can be classified into the following three categories, based on the core analysis process of dam siting: GIS/RS methods, MCDM/MCDM-GIS methods, and machine learning methods. The GIS/RS-based method has simpler data types and no complex factor weights to calculate. MCDM/MCDM-GIS-based methods are supported by powerful decision analysis systems that systematically deal with the weight relationships among diverse factors. Machine learning methods involve large amounts of data as training datasets. These three types of methods can be simplified into two categories: suitability-based evaluation and algorithmic model-based (Table 2). Based on the attributes of the factors, the types of factors involved in the decision-making process of dam siting are classified into six major categories: hydrology, geology, topography, water quality, environment, and socioeconomics. These six categories can be grouped into two categories based on the discipline area: environment and humanities–social sciences (Table 3). A total of 39 specific factors are involved in 6 major categories (Table 4).

**Table 2.** Categorization of dam siting methods.

| Classification | Specific Method |
|---|---|
| suitability-based evaluation | GIS/RS-based methods |
| | MCDM- and MCDM-GIS-based methods |
| algorithmic models-based | machine learning-based methods |

**Table 3.** Types of factors for dam siting.

| Classification | Specific Method |
|---|---|
| category of environment | topographical factors |
| | hydrological factors |
| | geological factors |
| | environmental factors |
| | water quality factors |
| category of humanities and social sciences | socioeconomic factors |

**Table 4.** A Listing of site selection impact factors.

| Criteria | Sub-Criteria | Percentage |
|---|---|---|
| topographical | slope | 88% |
| | elevation/hypsometry | 56% |
| | topographic wetness index (TWI) | 12% |
| | topographic position index (TPI) | 8% |
| | sediment transport index (STI) | 8% |
| | stream power index (SPI) | 8% |
| | terrain ruggedness index (TRI) | 8% |
| | plan curvature | 4% |
| | profile curvature | 4% |
| hydrological | rainfall/precipitation | 32% |
| | run-off/discharge | 32% |
| | drainage network order | 28% |
| | drainage density | 20% |
| | catchment size | 12% |
| | curve number grid | 12% |
| | stream width | 8% |
| geological | geology/lithology | 68% |
| | distance to faults | 28% |
| | distance to lineaments | 12% |
| | tectonic zones | 8% |
| environmental | land cover | 88% |
| | soil type | 52% |
| | distance to the streams/river | 32% |
| | groundwater | 32% |
| | soil erosion | 24% |
| water quality criteria | TDS | 16% |
| | electrical conductivity (EC) | 16% |
| | soluble sodium Percentage (SSP) | 12% |
| | potential of hydrogen (PH) | 12% |
| | sediment | 12% |
| socioeconomic | distance to roads | 32% |
| | distance to materials/facilities | 24% |
| | distance to cities/community | 20% |
| | distance to villages | 20% |
| | cost of construction | 8% |
| | welfare | 4% |
| | culture | 4% |
| | people incorporation | 4% |

## 3. Methods for Dam Siting

Dam siting methods can be divided into two categories: those based on suitability evaluation and those based on algorithmic models. The suitability assessment of dams is based on the use of geological, topographical, climatic, hydrological, socioeconomic, and other site selection factors, in order to determine the spatial extent of a suitable dam. Geographic information systems (GIS), remote sensing (RS), and multi-criteria decision-making (MCDM) methods are commonly used in the suitability assessment process, as they, by nature, are overlay analyses and have the advantage of being simple and easy to operate. These methods can be subdivided into single GIS/RS methods and MCDM-/MCDM-GIS integrated methods. Algorithmic model-based dam siting usually uses traditional machine learning algorithms, or the more popular fields of deep learning and artificial intelligence, to calculate the optimal location within existing spatial constraints. The results are more accurate compared to a suitability assessment, as the calculation process abstracts the data to points in spatial dimensions.

Finally, the three categories of methods are reviewed: GIS/RS methods, MCDM and MCDM-GIS methods, and machine learning methods.

### 3.1. GIS/RS-Based Siting

GIS is based on computer systems that process attribute data as well as spatial data, an effective tool for spatial analysis and decision making, with geographic information being a critical characteristic. RS provides an important way to acquire geographic information. Remote sensing products combine high temporal resolution with high spatial resolution, and are a critical source of data supporting spatial analysis. GIS, RS, and geospatial data play important roles in water resource management and allocation. While siting requires the collection and extraction of disparate data, and it is difficult and time-consuming to consider all criteria in the siting process, GIS can integrate multiple layers of data and provide an effective decision-making tool.

The process of GIS siting is based on supporting-decision rules that define how individual factors can be combined into ordered decision alternatives, based on assessment criteria. Decision rules in GIS use Boolean algorithms (Boolean) and weighted linear combinations (WLC) to classify factors into two categories: constrained and weighted. Boolean algorithms provide the "or" and "and" operators to impose constraints for factor retention or exclusion [33,34], while WLC calculates the total score by normalizing the score for each factor and, finally, weighting it [19,35–37]. If Boolean restrictions are described as hard or extreme, the WLC shows a more relaxed flexibility, with continuous suitability values. In a GIS analysis environment, WLC combined with Boolean algorithms are determined for dam siting options and involve a smaller number of factors [38–40]. Jamali [40] used Boolean logic in the siting of underground dams while combining fuzzy logic into a linear weighting process. The Boolean method limits the unsuitable part, then weights the factors of the remaining area to obtain three sites of high relative suitability and, finally, selects the best site in relation to the actual shape of the river valley.

In the analytical environment of GIS/RS, DEM (digital elevation model) is an important data in presenting topographic features. DEM has contributed greatly to the understanding of topographic features in the dam siting process and has served as an important input to calculate the drainage model [41]. UAV (unmanned aerial vehicle) photogrammetry is an advanced tool for acquiring high precision DEMs today, which has a low-cost advantage over light detection and ranging (LiDAR) while obtaining high resolution 3D topographic models [42]. Ajayi used UAV photogrammetry to obtain a high precision DEM, 3D model, and other by-products, and applied these to select suitable dam construction sites. The results show that UAV photogrammetry shows high applicability as a remote-sensing data source in the GIS analysis environment [43]. The high precision surface models produced by UVAs are more often used in the fields of dam monitoring [44,45], quantifying the effect of dams on sediment deposition [46–48]. Although the application to dam siting is not widespread, it provides a method and guide for dam siting.

### 3.2. MCDM- and MCDM-GIS-Based Siting

MCDM is a branch of the classical operations research model for solving trade-offs between the benefits of multiple conditions [49]. The process differs from the single GIS method by assessing the relative weight of each criterion, rather than making all factors equally weighted, and then comparing two or more alternatives [50]. Siting is an important type of decision making, weighing the importance of the elements in relation to each other in a given geo-spatial location and seeking the optimal site. MCDM is considered to be an optimization and improvement of simple logic of the Boolean and linear weighting type. It includes a range of sub-methods, which show their suitability and variability for different types of practical problems. Table 5 lists the main MCDM methods, including authors and time, where analytical hierarchy process (AHP), Elimination and Choice Translating Reality (ELECTRE) and Technique for Order Preference by Similarity to an Ideal Solution (TOPSIS) are commonly used for dam siting.

**Table 5.** Several methods in MCDM.

| Name | Full Name | Primary Author | Time | Ref. |
|------|-----------|----------------|------|------|
| ELECTRE | Elimination and Choice Translating Reality | Benayoun R. | 1966 | [51] |
| AHP | analytical hierarchy process | Saaty T.L. | 1970s | [52] |
| DEMATEL | Decision Making Trial and Evaluation Laboratory | Gabus A., Fontela E. | 1972 | [53] |
| TOPSIS | Technique for Order Preference by Similarity to an Ideal Solution | Hwang C., Yoon K. | 1981 | [54] |
| PROMETHEE | Preference Ranking Organization Method for Enrichment Evaluations | Brans J. P., Vincke P. | 1984 | [55] |
| ANP | Analytic Network Process | Saaty T.L. | 1996 | [56] |
| VIKOR | Vlsekriterijumska Optimizacija I Kompromisno Resenje | Opricovic S. | 1998 | [57] |

Decision analysis involves a system for assessing complex problems, where integrating MCDM into GIS can improve analytical capabilities. Jacek [58] stated that MCDM-GIS is an important branch of geographic information science that promotes the two main fields of spatial decision support and participative GIS. GIS has advantages in the multidimensional processing of geographic information, and MCDM is used to compare alternatives. As decision problems become more complex, a single MCDM tends to combine the advantages of GIS in geo-analytical processing, which has been used for dam siting in several countries: China [59], Iran [17], Iraq [60], Nigeria [41], Kenya [61], India [62], Saudi Arabia [63], Pakistan [64], and so on.

AHP and TOPSIS have been very widely used in practice. Noori [60] compared the accuracy of two methods—AHP and fuzzy logic—by selecting 12 condition factors in 5 categories: hydrology, topography, geology, environment, and socioeconomics. The accuracy of the two methods was also evaluated, based on the government's proposed dam site data. The results of the study showed that the results produced by the AHP method were more dispersed throughout the study area; conversely, those of the fuzzy model were more clustered. In terms of overall accuracy, fuzzy logic is more accurate than AHP for conducting dam site suitability assessments. Noori [65] also combined TOPSIS with fuzzy logic to obtain the optimal decision solution by calculating weights for 18 factors under social, economic, and environmental constraints. Jozaghi [17] considered AHP and TOPSIS as the most widely adopted methods for solving water resource problems. He used TOPSIS and AHP with 11 factors to compare the two methods. It was concluded that the TOPSIS method is more appropriate, in this setting, to address the siting of dams. Building dams to store water can mitigate the effects of drought and ensure water supply, under the background of the significant reduction in annual precipitation in the Middle Eastern country of Iraq. Othman [19] compared the AHP with the linear weighting method (WSM), selecting 14 factors as inputs to the model. The results were compared with the 21 pre-selected dam locations suggested by the Ministry of Agriculture and Water Resources (MAWR), showing that the overall accuracy of the AHP was significantly better than that of the WSM.

*3.3. Machine Learning-Based Siting*

In addition to relatively traditional MCDM methods, there has been a growing trend among scholars of using machine learning to solve a range of decision-making problems. Simon [66], the originator of Simon's decision theory, argued that the decision domain is weakly structured, knowledge-rich, and non-quantitative in character. With the rapid development of machine learning, the application of artificial intelligence and machine learning to the decision-making domain can overcome the shortcomings of existing decision-making solutions. Intelligent decision support systems (iDSS) [67] is another sub-discipline in decision support systems research, developed from the interaction of decision targets and machine learning. Machine learning is the kernel of the system, making decision support systems more adaptive, adapting to different decision makers, and changing environments through the constant acquisition of new content.

Machine learning has been used, in siting decisions, to process large and complex data, as a complement to MCDM, or as a complement to each. Rami Al-Ruzouq [68]

combined GIS, MCDM, and machine learning techniques to generate suitability maps, in order to determine dam sites. Three machine learning methods were used and compared: support vector machine (SVM), random forest (RF), and gradient augmented tree. The results showed that RF had the highest accuracy. The highest accuracy results for RF were similarly obtained by Pourghasemi [69], who compared five machine learning models—SVM, RF, boosted regression trees, mixture discriminant analysis, and multivariate adaptive regression spline—and selected 14 environmental factors to generate dam suitability maps. The results showed that distance to the river and drainage density were the two most important environmental factors. All five different machine learning methods had high accuracy, with RF having the highest accuracy.

Machine learning applications for dam siting are still evolving, and more research has been done using classical machine learning models. Even with the development of deep learning and artificial intelligence, applications in this area are still rare. Nevertheless, machine learning has been widely used in water resource and environmental studies, such as groundwater level mapping and potential analysis [70], flood sensitivity analysis and prediction [71–73], and landslide sensitivity analysis and prediction [74–84].

## 4. Factors Influencing Dam Siting

Factors are important elements in the siting process and affect the outcome of the site from different perspectives. A review of a large number of studies found that the selection of factors between dams showed certain similarities and characteristics, considering the differences that exist in the natural environment, social environment, and purpose. Analyzing the types and frequencies of factors in different articles can be useful in providing a reference for future site selection studies. It is essential to have advanced knowledge regarding the use of current study factors. For example, Othman [19] analyzed the factors in different papers before making a decision on the selection of factors, and concluded that 70% used land use, soil type, slope, sedimentation, and CN grid; 20–40% used elevation, drainage networks, distance to lineaments, lithology, distance to faults, tectonic zone, distance to villages, distance to roads, and distance to towns; and less than 10% of the articles used distance to materials, total dissolved solids (TDS), evapotranspiration, and depression volume. Nevertheless, rainfall, slope, land-use, geological lithology, and soil type are all important factors in different siting scenarios.

### 4.1. Criteria for Dam Siting

In this section, the 39 site selection criteria from the 25 sample papers are assessed and grouped into six categories: hydrological, geological, topographical, water quality, environmental, and socioeconomic. The first five categories of factors belong to the broad range of environmental influences, while the last—socioeconomic—falls under the category of humanities and social sciences. Table 4 presents the frequency with which different siting criteria were chosen. We also discuss the differences in criteria due to the various purposes of dams.

Topographic factors reflect important topographic features that directly determine whether a dam can find a suitable or optimal location. The slope and elevation rates are 88% and 56%. The hydrological factors characterize the potential for abundance or scarcity of water resources in the target area. Rainfall and runoff are cited with 32% probability and drainage network order with 28%, where runoff is a reflection of a fraction of the rainfall. Among the geological factors, geology/lithology is used at 68%. The rates of land use and soil type in environmental factors were 88% and 52%, respectively. The probability of water quality criteria as a weighting factor is low. Distance to roads was the most important in the economic community factors, with 32%. The quantitative analysis of the sample literature provides guidance for the selection of factors for future dam siting research. These factors are elaborated in the following sub-section.

### 4.1.1. Topographical Factors

Elevation and slope are the main criteria reflecting topographic characteristics. It is generally accepted that areas of moderate elevation are more suitable for dam construction, while lower and higher elevations show weak suitability [19,60]; however, researchers differ in their views on the suitability of steep versus moderate slopes for dam construction.

Othman [19] argued that steepness is the main factor influencing dam siting, with smooth land being more suitable for dam construction than steep slopes, as did Buraihi [85]. As the slope increases, so does the risk of landslides and the pressure on building foundations [59,86]. Ruzouq [68] concluded that water velocity is proportional to slope, and that a slope of less than 5% has a positive effect on soil and water conservation in reservoirs, while Jha [87] argued that the slope should be less than 15°. Groundwater dams are also chosen to be built on a gentle slope in wide valleys [64]. On the other hand, Wimmer [88] argued that excellent locations for dams are often found in canyons with steep slopes, where the topography can be used to the advantage of a shorter dam axis length and a larger capacity can be achieved with less construction earthwork, in agreement with Teschemacher [89]. Jozaghi [17] suggested that river valley shape is critical and that dams are preferable to be built at narrow locations where the upper river valley opens up, and that slope is generally greater at these locations. Becue [90] relates valley morphology to dam site and types of dams. Natural structures with ideal river valley morphology are rare, so different dam types are fitted to different river valley morphologies. Wide valleys are suitable for earthfill dams, narrow sites for gravity dams, and even narrower sites for arch dams. The suitability of the slope is also closely related to the purpose of the dam: when used for rainwater harvesting, the Food and Agriculture Organization (FAO) recommends that the slope is no greater than 5% [91]. In the siting of check dams, the slope determines the reservoir capacity and sedimentation, where a greater slope leads to greater sedimentation.

TWI describes the spatial pattern (location and size) of saturated areas affected by watershed-scale hydrologic processes, and characterizes the proportional relationship between moisture and contributing areas [92]. TPI represents the difference between the raster and the mean elevation of the neighborhood (within the neighborhood), and is used to understand the relationship between runoff generation, flow rates, and sediment transport [93,94]. STI considers the influence of topography on erosion and clarifies the sediment transport potential to characterize erosion and sedimentation processes [95]. SPI is used to quantify the erosive capacity of rivers, where the flow is a determinant of channel erosion and flood damage [96]. TRI is one of the main factors affecting river potential energy, surface water storage capacity, runoff velocity, and river path at the watershed scale, characterizing elevation differences between adjacent raster [97]. Rahmati [98] proposed a check dam siting for soil and water conservation purposes, highlighting the importance of hydro-topographic factors and using five indices: TWI, TPI, STI, SPI, and TRI. Jamali [64] used TWI to quantify the influence of hydrological processes by topography in groundwater siting in Pakistan.

Ildoromi [99] chose three criteria in terrain characterization, in addition to slope, which are the most commonly used. Plan curvature and profile curvature were also used to express topographic features, which play an important role in embodying erosion and sedimentary geomorphology.

### 4.1.2. Hydrological Factors

Hydrological parameters determine the catchment capacity of the target area. The main hydrological factors include rainfall, runoff, catchment size, river network level, river network density, river width, and drainage network order. The statistics revealed that rainfall, runoff, river network class, and river network density were used more than 20% of the time, in order to provide reference for future studies.

Precipitation is the main source of runoff recharge, which has a positive impact on dam function when there are no natural disasters caused by extreme heavy precipitation events (e.g., landslides or floods). The runoff water quantity volume can be evaluated

using the CSC-CN hydrologic process model [100], or roughly estimated by the Hydrologic Analysis module of ArcGIS [101]. The runoff curve number (CN) is an important tool for evaluating runoff volume. The CN value depends on soil type, land-use and land-cover, and hydrogeological conditions. The size of the catchment area is a prerequisite for determining the location of the dam. The catchment area should be large enough to maintain average reservoir storage, but not so large that the volume of water is often in excess of the reservoir capacity; otherwise, a more costly spillway would need to be built. The river network provides the essential runoff for the dams, and different river network classes indicate different runoff volumes when the rivers are upstream tributaries and downstream mainstems. River network density reflects the water resources in the region, and a higher river network density shows better diversion capacity in the face of floods, while river network density and flood volume show a positive correlation trend [98]. The river network rank indirectly reflects the runoff volume, where higher-order rivers have higher runoff volume.

### 4.1.3. Geological Factors

The geological conditions of the dam site are critical and directly affect the safety and stability of the project. The geological foundation of the site also affects the dam type [102] and dam construction materials [86,103]. The site should have impermeable geology, dam foundation, and no leakage; for example, southwest China is a typical karst landscape region, and the lithology directly affects whether the water will "leak away" after the dam is built [104]. Geological-related indicators include geology/lithology, tectonic zones, distance to faults, and distance to lineaments.

Lithology is the most important geological factor [17], which was used 68% of the time, with which the influence of faults and tectonic lines are considered. Different epochs form ground rock units representing different conditions of stability and degreez of pressure resistance [19]. Karst areas, due to the karst phenomenon of carbonate rocks, have been observed in China, where the construction of underground dams in karst areas has led to excessive groundwater siltation which, in turn, has led to flooding in the upstream lowlands [105]. Therefore, the construction of underground dams needs to consider the water retention capacity of karst areas and the complexity of the underground routes of caves. Faulting is one of the main factors causing landslides [106]; thus, the farther the site is from a fault, the lower the risk of landslides. Unstable tectonic zones and lineaments also have potential risks.

While 60–70% of the analyzed papers chose 1 or 2 geological factors, Othman [19] chose four geological factors: tectonic zones, lithology, distance to lineaments, and distance to faults. Less suitable tectonic zones include the Imbricated Zone and the High Folded Zone, where faults and lineaments represent geological weaknesses and are usually avoided by buffer zones.

### 4.1.4. Environmental Factors

The environment is a broad concept. In this case, we chose environmental factors including soil environment (soil type, erosion), land-use, distance to water resources, and groundwater resources. However, this is obviously not representative of all environmental criteria, and are only common environmental factors used in the sample literature. The two criteria of land-use and soil type had the highest usage rates: 88% and 52%, respectively.

Soil types can be classified according to soil texture, which leads to different rates of soil infiltration and, thus, different effects on the runoff volume. Sufficiently water-resistant fine-grained foundations, clays, and clay mixtures are recommended [102,107]. High population activity, increased construction, and deforestation are the main causes of soil erosion [108–110], and increased erosion in the watershed has a direct impact on the accumulation of sediment in the reservoir, making areas of soil erosion less suitable for dams intended for water storage. Land-cover reveals land-use patterns, as well as links to current social development. Changes in land-use and vegetation usually affect the water

cycle [111]. Furthermore, the different land-use types can be quickly filtered, in order to identify areas suitable for development and use. The distance to water resources reflects the connection to the river network in that watershed and, if the dam is still some distance from the river, then water resources may become the biggest problem facing it.

### 4.1.5. Water Quality Factors

Water quality criteria are mostly considered for the siting of groundwater dams, which determine the quality and drinking safety of groundwater resources for development and utilization, however, surface dams for irrigation or water supply purposes are among those that use water quality factors.

Jozaghi [17] proposed indicators of salinity and sodium levels of water bodies to limit the water quality conditions used for agricultural irrigation: TDS, SSP, PH, and EC. These indicators are related to agricultural needs, such as soil suitability for cultivation, infiltration rate, and plant growth suitability. Ruzouq [68] also used the TDS to characterize the quality of water in a study area located in the Arabian Gulf, where the high salinity of the water is the main problem that must be overcome and avoided.

### 4.1.6. Socioeconomic Factors

Different social settings manifest as differences in socioeconomic criteria for siting. Closeness to roads and settlements, leading to lower transportation costs, had a 32% usage rate. Distance to material facilities, roads, cities, and villages are used to quantify construction cost issues. Distance to the countryside and distance to the city are two different situations, which usually require a certain distance from the city but being as close to the countryside as possible. This is because rural areas can bring the required labor force, while dams are not suitable within a certain buffer zone within the city, in order to prevent large accidents such as dam failures [112].

Othman [19] chose the distances from the city, the countryside, and the road to weigh the optimal solution, in terms of economic cost. Emamgholi [106] presented unique social factors—the welfare, culture, and participation of the residents in the actual work—and conducted field investigations. Dam construction has an impact on the nearby residents, and the support of local residents determines the smoothness of the project.

### 4.2. *The Influence of Dam Use on Criteria Selection*

The objectives of decision makers vary widely, making it difficult to generalize a number of criteria for dam siting. Decision factors vary by purpose of the dam, from large hydroelectric power generation dams (e.g., the world-leading Three Gorges Dam) to small dams for irrigation and aquaculture.

Lempérière [113] considered the dams of the future as being multipurpose, while Abushandi [63] identified five major purposes of modern hydraulic facilities: regulation and flood control under an extremely uneven spatial distribution of water resources, soil erosion and sediment control, drought control, irrigation, and hydropower generation. According to the latest data from the ICOLD 2020 statistics (Table 6) [16], irrigation is the major purpose, accounting for 47% and 24% of the sole-purpose and multiple-purpose statistics of dams, respectively. The next three major purposes are hydropower, water supply, and flood control.

**Table 6.** Purposes of dams.

| Description | Sole-Purpose | Percentage | Multiple-Purpose | Percentage |
|---|---|---|---|---|
| flood control | 2539 | 8.82% | 4911 | 0.19% |
| fish farming | 42 | 0.15% | 1487 | 0.06% |
| hydropower | 6115 | 21.24% | 4135 | 0.16% |
| irrigation | 13580 | 47.17% | 6278 | 0.24% |
| navigation | 96 | 0.33% | 579 | 0.02% |
| recreation | 1361 | 4.73% | 3035 | 0.11% |
| water supply | 3376 | 11.73% | 4587 | 0.17% |
| tailing | 103 | 0.36% | 12 | 0 |
| others | 1579 | 5.48% | 1385 | 0.05% |

Source from [16].

The purpose determines the siting of different water collection structures and hydraulic facilities, such as retention basins, reservoirs, check dams, and rainwater harvesting structures (RWH), in order to achieve the spatial regulation of water resources. Check dams are built on seasonal streams, in order to intercept runoff from catchment-contributing areas and store it to optimize water utilization. RWH are important technologies for storing fresh water or recharging groundwater resources, for purposes such as water supply and agricultural irrigation. Singh [114] pointed out the differences between four types of catchment structures—RWH, check dams, percolation tanks, and farm ponds—in the process of determining sites (Table 7).

**Table 7.** Siting differences of catchment structures.

| Type | Slope | Permeability | Land Use | Soil |
|---|---|---|---|---|
| RWH | <15% | low | near agricultural land | silt loam |
| check dams | <15% | less | barren, shrub, riverbed | sandy clay loam |
| percolation tank | <10% | high | barren, shrub | silt loam |
| farm ponds | <10% | moderate | barren, shrub | sandy clay loam |

Source organized from [114].

In order to clarify the link between the siting factors and purposes of dams, information was collated from 25 sample papers, selected for the four main types of dam purposes: irrigation, hydropower, water supply, and flood control. Table 8 reflects the frequencies (in percentage) of criteria used in different types of dams, where the total percentages of sub-criteria under each type of criteria in the four types of uses is 100%.

The preferred factors in topography are slope and elevation, which highly influence the construction of irrigation, hydropower, and flood control dams, and which together account for 21%, 23%, and 19% of these three types of uses, respectively. Runoff and rainfall are important hydrological factors, which are more important in irrigation, water supply, and flood control, accounting for 17%, 12%, and 15% respectively. Geological factors are significantly more prominent in hydropower dams than for the other three purposes, up to 22%. The most important of the environmental factors are land-use and soil type, with higher percentages for irrigation and water supply, accounting for 30% and 15% respectively. Water quality indicators are concentrated in dam siting studies for water supply and irrigation purposes, hydropower and flood control types of dams are usually not involved in water quality standards. Finally, socioeconomic factors maintained relative importance in all purposes.

**Table 8.** Share of criteria in different purposes.

| Criteria | Sub-Criteria | Irrigation | Hydropower | Water Supply | Flood Control | Total |
|---|---|---|---|---|---|---|
| topographical | slope | 9% | 15% | 6% | 16% | 46% |
| | elevation/hypsometry | 12% | 8% | 2% | 3% | 25% |
| | TWI | 7% | - | - | - | 7% |
| | TPI | - | - | 4% | - | 4% |
| | STI | - | - | 4% | - | 4% |
| | SPI | - | - | 4% | - | 4% |
| | TRI | - | - | 4% | - | 4% |
| | plan curvature | - | - | 3% | - | 3% |
| | profile curvature | - | - | 3% | - | 3% |
| hydrological | rainfall/precipitation | 8% | 2% | 4% | 9% | 23% |
| | run-off/discharge | 9% | - | 8% | 6% | 23% |
| | drainage network order | 5% | 2% | 5% | 6% | 18% |
| | drainage density | 4% | 2% | 8% | - | 14% |
| | catchment size | - | 5% | 4% | - | 9% |
| | curve number grid | 3% | - | 5% | - | 8% |
| | stream width | - | - | 3% | 2% | 5% |
| geological | geology/lithology | 9% | 26% | 6% | 16% | 57% |
| | distance to faults | - | 18% | - | 6% | 24% |
| | distance to lineaments | - | 9% | - | 4% | 13% |
| | tectonic zones | - | 6% | - | - | 6% |
| environmental | land cover | 12% | 8% | 10% | 10% | 40% |
| | soil type | 18% | - | 5% | - | 23% |
| | distance to the streams/river | 6% | - | 8% | - | 14% |
| | groundwater | 5% | - | 12% | - | 17% |
| | soil erosion | 6% | - | - | - | 6% |
| water quality criteria | TDS | 16% | - | 16% | - | 32% |
| | EC | 8% | - | 9% | - | 17% |
| | SSP | 8% | - | 9% | - | 17% |
| | PH | 8% | - | 9% | - | 17% |
| | sediment | 8% | - | 9% | - | 17% |
| socioeconomic | distance to roads | 14% | 2% | 8% | 4% | 28% |
| | distance to materials/facilities | - | 12% | - | 5% | 17% |
| | distance to cities/community | 8% | 6% | - | 4% | 18% |
| | distance to villages | 3% | - | 12% | 3% | 18% |
| | cost of construction | - | 6% | - | 3% | 9% |
| | welfare | - | 2% | - | - | 2% |
| | culture | - | 2% | - | - | 2% |
| | people incorporation | - | 6% | - | - | 6% |

In the irrigation of crops and domestic water supply, water quality standards are important factors in the siting of surface and underground dams and rainwater harvesting structures [17,115]. To ensure crop safety and food security, water quality standards are important factors in dams for irrigation and water supply purposes. Poor water quality can negatively affect crop productivity, crop quality, and the public health of consumers and farmers, who are in direct contact with the irrigation water [116]. Globally, at least 20 million hectares of agricultural land are irrigated with treated or untreated wastewater [117], often containing excess sodium, magnesium, chloride, and boron, which affect soil alkalinity, phytotoxicity, and heavy metal content. However, geological factors play a dominant role in underground dams for such purposes, including sub-factors such as geological lithology, distance to faults, and distance to lineaments [118–120].

Hydroelectric power plants are dams designed to generate electricity by impounding rivers and converting the kinetic and potential energy of water into electrical energy using hydraulic turbines. According to the ICOLD, there will be 6115 dams for the purpose of power generation by 2020, of which 4135 will be multi-purpose dams [16]. The Three Gorges Dam, one of the world's 10 largest dams, is a multi-purpose dam that not only provides a huge supply of electricity, but also provides excellent flood control [121]. The hydrological index [122], installed hydroelectric capacity [123], and potential power generation [124] are the main considerations in the siting design of hydropower dams. Rojanamon [125] proposed four directions of consideration for the siting factors of power

stations—engineering, economic, environmental standards, and social impacts—and integrated the sub-factors of each direction, using GIS analysis to process to obtain the best potential siting area. Jafar [126], on the other hand, based on GIS and combined with the best-worst method (BWM) in MCDM, determined the optimal location model for hydropower dams, in terms of four aspects: physical, environmental, socioeconomic, and technological.

Floods and other water-related disasters account for 70% of all deaths associated with natural hazards [127], and flood control is one of the most important elements of sustainable water resource management [128]. Flood control dams can largely mitigate the catastrophic effects of floods. There are 2539 sole-purpose flood control dams and 4911 multi-purpose flood control dams worldwide [16]. In Egypt, which suffers from frequent seasonal flooding and droughts, as well as water demand for agricultural irrigation, the Aswan Dam largely regulates the extremely uneven distribution of water resources and achieves spatial and temporal deployment of the multi-year runoff from the Nile [129]. The critical factors for the siting of flood control dams include the design height of the dam, which is limited by topographic conditions, hydrological characteristics, and technology, where the height of the dam directly affects the possibility of flooding and, indirectly, the possibility of dam failure [130]. Sumi [131] also pointed out that the relationship between dam height and storage capacity varies greatly between countries, due to differences in geographic conditions; for example, the ratio of storage capacity to dam height is particularly large for dams in the United States, as these dams are often built in gently sloping rivers and wide river valleys. Patel [132] considered the good soil and water conservation functions of check dams to moderate flooding and soil erosion in small watersheds.

## 5. Conclusions

A study of the literature related to dam siting from 2000 to 2020 was carried out, considering the determination of dam siting suitability and design techniques under different scenarios in practice, in terms of siting methods, siting variability of different dam types, and factors affecting the siting process.

In terms of siting methods, GIS is the foundation of dam siting, being an important data analysis platform that is difficult to completely disconnect from the entire siting process. GIS provides powerful computing power to achieve multi-layer data integration and result analysis, which has shown excellent capabilities for visualization and data management. With the complexity of site selection requirements and the large amount of spatial data associated with various technical, economic, social, and environmental criteria, it is difficult to identify the sites through GIS alone, such that the integration of MCDM models in the GIS environment is a necessity for development. In addition, the development of UAV provides direction for dam siting, and higher precision data sources are a strong support for geospatial analysis. Machine learning, on the other hand, has shown better efficiency, as well as higher accuracy when dealing with large and complex data sources. MCDM remains the dominant site selection decision method, however, the ability to cope with more complex and variable decision processes in the future allows for a broader application of machine learning.

Various factors play their respective roles in dam siting. Dams have different purposes and uses, and the differences in use are related to siting factors. Water quality and sedimentation are of greater concern in irrigation and water supply, which affects the safety of crops and drinking water; flood control is more concerned with topographic conditions, hydrologic characteristics, and technical capabilities; and, for power generation, the most important factor is the conversion of maximum dynamic and potential energy to electrical energy from water energy, as mapped by hydrologic elements. Dam function influences these factors, and these factors determine the suitability of design of the dam, thus complementing each other, with accuracy and rationality of site selection being the ultimate goal.

Existing site selection methods also have irrationalities and shortcomings, such as the scientific and transparent nature of the scoring process, the consideration of alternatives in the decision-making process, and the stakeholders involved. The literature lacks content on whether the results were adopted by the function and whether the multidisciplinary analytical context was considered. In addition, bottom-up approaches are often difficult to implement, with the choice of criteria and factors usually made by the authors of the paper, or based on a literature review or expert consultation. Promoting multi-actor participation is not easy, considering the divergence of different stakeholders in perspectives and values. All decisions should also have some top-down thinking, with links established with line ministries and government agencies prior to decision making, in order to advance the implementation of dam siting plans.

Dam siting is the study of site selection, a branch of decision making, which has the characteristics of multidisciplinary integration, involving decision making, and coordination, geographic information science, computer science, etc. Siting decisions are constantly iterated and updated as the discipline evolves, and the dam siting process will inevitably face more challenges.

## 6. Discussion

### 6.1. Method Evolution Trend

Our reviews of nearly 20 years of siting research demonstrated iterative advances in dam siting methods. From a single GIS-based approach, it has evolved to an integrated GIS-MCDM approaches, or machine learning technique-driven approaches. The pursuit of precision and accuracy in dam siting has led to the refinement and improvement of methods. Machine learning is an important development prospect for site selection, and the existing research has made extensive use of traditional machine learning models, such as SVM, RF, and decision trees. The application of deep learning and neural networks in this field is very immature, thus offering ample possibility for future development. The factors in dam siting tend to be more complex and, as such, simple models show limitations. Therefore, the integration of methods is also an important development direction for future siting research, in order to achieve complementary advantages. Chowdhury [133] also introduced a gamma algorithm into fuzzy AHP to find the best catchment in his latest study, constituting a bold attempt to integrate methods.

### 6.2. Outlook for Dam Purposes

From the ICOLD statistics, it is found that irrigation, power generation, and flood control are the most dominant dam uses. In Section 4.1, a quantitative analysis of factor selection for the four categories of use—irrigation, hydropower, water supply, and flood control—is presented, with different uses reflecting preferences for factor selection. According to the ICOLD, in addition to these four categories of use, recreation dam purposes, although insufficient in number, have more room for development, in line with the context of the times. From 5% in 2014 [63], the share of such dams has risen to 11% by 2020. With the rapid economic development in China, it has started to reflect on the serious environmental problems, where the protection and management of ecology is an important part of sustainable development. At the same time, China's urbanization rate has reached 45.4%, while the ensuing urban diseases are in urgent need of solution. With ecology as the origin, combined with the recreational purposes of dams, the building of water-rich structures in cities has been recommended; the ecological benefits are enhanced, while providing recreation for residents. New types of dams designed with this concept in mind are expected to become a future development trend.

### 6.3. Seeking Optimal Allocation of Factors

It is critical that the purpose of a dam be closely linked to its associated factors. The lack of consideration of dam siting factors can lead to a number of consequences. Opuha Dam was designed for water storage, hydroelectric power generation, and increased

summer-dry water flow; however, the failure to consider hydrological conditions and aquatic environment led to algal blooms without sufficient water flow to flush the riverbed to remove the attached organisms [134]. Further, new dams should focus more on the relationships between dam purposes and different factors, such as assessing the synergies between the distribution of biophysical, socioeconomic, and geopolitical impacts [135].

Among all the factors, topographic factors are the most basic prerequisite for finding a suitable location. Slope, elevation, sink flow, and valley shape are crucial for evaluating suitable sites. Many studies have started from topographic features and designed programs to automatically obtain target dam locations [88,89,136]. The upstream location of dams is usually depressed terrain. Thus, the judgment of suitable sites for a dam can be transformed into the identification of depression topography; however, the concept of a depression is relative. An uneven concrete road can be considered as a collection of numerous depressions, and the Tarim and Sichuan Basins can also be considered depressions, when looking at China on a national scale. This inspires the importance of scale for terrain feature identification, and proposes the extraction of depressions and dam siting based on multi-scale terrain features.

Figure 2 illustrates the process of finding a suitable site by transforming the raster size. The search for the optimal location must satisfy the following conditions: (1) located at a canyon pass, (2) high rate of elevation change; and (3) concave terrain or saddle. By setting different raster sizes, the higher the elevation change rate, the higher the average elevation value of the raster. Compared to the interior of a depression with gentle topography, a local raster with the highest mean elevation value is the optimal solution for the site.

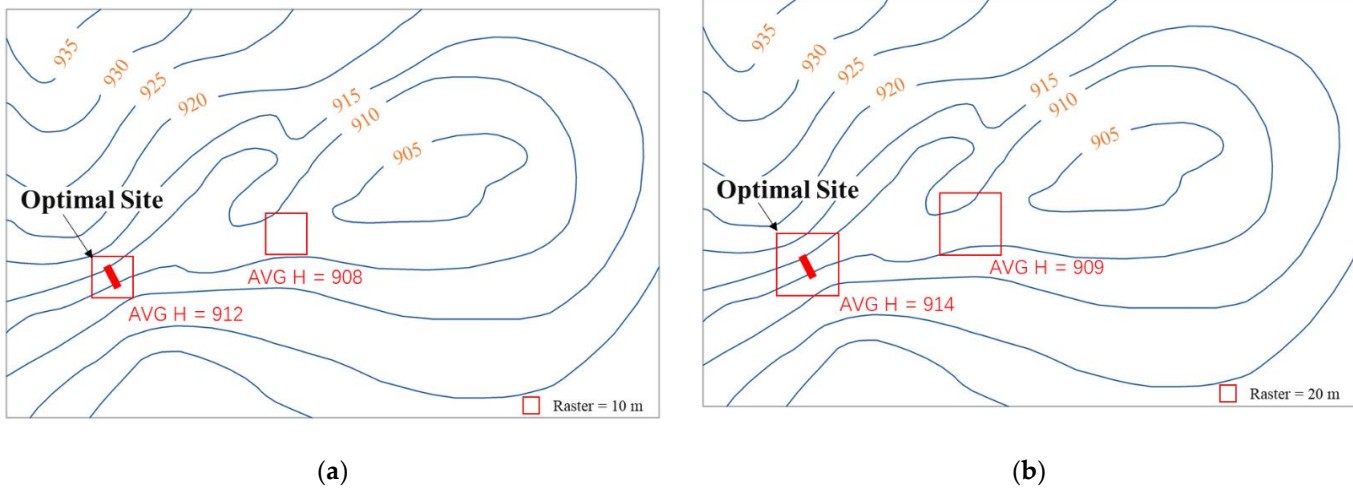

|     |     |
| :-: | :-: |
| (**a**) | (**b**) |

**Figure 2.** Multi-scale terrain feature-based depression extraction: (**a**) when the raster is 10 m; (**b**) when the raster is 20 m. AVG H = average height.

### 6.4. Public Participation

The promotion of public participation in the decision-making process should be emphasized. Public participation is weak in existing siting factors. The site selection process is a decision-making process, which has evolved from a "closed" expert-oriented process to an "open" user-oriented process [58]. Dam construction can have certain negative impacts on the nearby environment and citizens, and decision makers should be aware of the important role of public participation, which is conducive to an open, transparent, and democratic process. When the dam construction process lacks public participation links, the siting process lacks social engagement; however, public involvement faces a series of problems such as difficulty of participation, lengthy times, high cost, and disagreement, where the coordination of these issues is the main challenge. In this process of public participation, the concerns, needs, and values of citizens must be integrated into the

decision-making process by government and authorities, forming an active process of two-way communication.

**Author Contributions:** Conceptualization, Y.T. and Y.W.; methodology, Y.T. and Y.W.; formal analysis, Y.W.; writing—original draft preparation, Y.W.; writing—review and editing, Y.T., Y.C. All authors have read and agreed to the published version of the manuscript.

**Funding:** This research was funded by the State Key Program of National Social Science Foundation of China, No. 18AJY018.

**Institutional Review Board Statement:** Not applicable.

**Informed Consent Statement:** Not applicable.

**Data Availability Statement:** No new data were created or analyzed in this study. Data sharing is not applicable to this article.

**Acknowledgments:** In this study, the papers were downloaded from the Web of Science and Scopus, thanks to the retrieval and download services of the above websites. Thanks also to the statistics provided by ICOLD. The authors are grateful to the anonymous reviewers for their constructive and crucial comments that helped improve the manuscript. The authors also thank supervisors and classmates for their help and support during the research.

**Conflicts of Interest:** The authors declare no conflict of interest.

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
