# Peer review of "Dam Siting: A Review"

_water, doi:10.3390/w13152080_

Round 1

Reviewer 1 Report

Dear authors,

The present paper aims to highlight the factors that condition the selection of the appropriate location for the construction of dams and the most appropriate siting methods. Hydrological, geological, topographical, water quality, environmental, socio-economic factors and parameters are covered in a convincing way.

It is an interesting work and potentially publishable, with obvious practical implications in water resource management.

I recommend publication of this work with some minor considerations.

In my opinion, there are some aspects that could be revised, so that the manuscript should be more consistent.

I do not consider relevant the aspects related to mentioning the databases used to find in literature the issues of the location of the dams or of the journals in which the publication was made. Thus, I suggest giving up subsection 2: Investigation and Analysis of Related Literature.

Regarding the part on the use of remote sensing and GIS, references regarding the applicability of UAV photogrammetry and 3D mapping (3D surface models) could be added. The DEMs generated from the images taken from the drones have high resolution and can be used successfully to determine the most suitable location of the dams using topographic constraints. The digital elevation model represent an input to calculate the drainage model.

As for the bibliography, not all references follow exactly the editing rules contained in the author's guide.

I encourage the authors to work to improve their manuscript.

Reviewer 2 Report

Manuscript presents a state of the art review review on Dam Siting. Topic is of relevance and high impact. Several major revisions must be carefully considered before acceptance.

+ Former reviewer papers on Dam Siting must be reviewed and the diference and originality of this paper must be discussed in the introduction.

+ Review methodology is not discussed; how the papers were identified? how PRISMA was implemented? how the screening were done?

+ review needs a flowchart or graph

+ The research taxonomy is not given. Please provide a taxonomy of search in the form of a flowchart or a graph

+ How the methods of the reviewed papers are evaluated? a comparative analysis of the papers must be given

+ now that you collected a group of paper, then what? what had been concluded? a valid lessons must be learned. Discuss it after the results.

+ What is the future research, trend, past achivement on Dam Siting. Elaborate more.

+several claims are not cited. Some figures are not cited or adapted correctly.

Round 2

Reviewer 1 Report

The authors responded carefully to the reviewers' comments and suggestions, modifying or completing the article accordingly. In this way the study gained more consistency, it is quite clear and expressive enough.

I recommend publishing the article in its current form.

Reviewer 2 Report

My comments had been addressed. I suggest acceptance.